# A Bayesian Network Meta-Analysis and Systematic Review of Guidance Techniques in Botulinum Toxin Injections and Their Hierarchy in the Treatment of Limb Spasticity

**DOI:** 10.3390/toxins15040256

**Published:** 2023-03-31

**Authors:** Evridiki Asimakidou, Christos Sidiropoulos

**Affiliations:** 1Klinik Hirslanden, Witellikerstrasse 40, 8032 Zürich, Switzerland; 2Department of Neurology, Michigan State University, East Lansing, MI 48824-7015, USA

**Keywords:** botulinum neurotoxin, injections, limb spasticity, electrostimulation, electromyography, ultrasound, anatomical localization

## Abstract

Accurate targeting of overactive muscles is fundamental for successful botulinum neurotoxin (BoNT) injections in the treatment of spasticity. The necessity of instrumented guidance and the superiority of one or more guidance techniques are ambiguous. Here, we sought to investigate if guided BoNT injections lead to a better clinical outcome in adults with limb spasticity compared to non-guided injections. We also aimed to elucidate the hierarchy of common guidance techniques including electromyography, electrostimulation, manual needle placement and ultrasound. To this end, we conducted a Bayesian network meta-analysis and systematic review with 245 patients using the MetaInsight software, R and the Cochrane Review Manager. Our study provided, for the first time, quantitative evidence supporting the superiority of guided BoNT injections over the non-guided ones. The hierarchy comprised ultrasound on the first level, electrostimulation on the second, electromyography on the third and manual needle placement on the last level. The difference between ultrasound and electrostimulation was minor and, thus, appropriate contextualization is essential for decision making. Taken together, guided BoNT injections based on ultrasound and electrostimulation performed by experienced practitioners lead to a better clinical outcome within the first month post-injection in adults with limb spasticity. In the present study, ultrasound performed slightly better, but large-scale trials should shed more light on which modality is superior.

## 1. Introduction

Botulinum neurotoxin (BoNT) chemodenervation is widely used in the treatment of neurological disorders characterized by muscle overactivity, such as dystonia and spasticity [1,2,3]. BoNT acts at the neuromuscular junction by hindering the release of acetylcholine from presynaptic neurons [4,5,6]. This leads to a subsequent blockade of neurotransmission and flaccid paralysis of the muscle [4,5,6]. An in-depth knowledge of anatomy and manual skills are a prerequisite for successful implementation of intramuscular BoNT injections, which can be particularly challenging in certain anatomical regions [7,8,9,10].

The current study focused on limb spasticity. Spasticity is a condition characterized by increased muscle tone, stiffness and exaggeration of stretch reflexes, which can lead to restrictions in movements and speech production, discomfort and pain [11,12,13]. Accurate targeting of spastic muscles is fundamental. A precise anatomical localization maximizes the amount of BoNT administered within the target structure, at the same time minimizing unintended weakness of adjacent muscles due to toxin spread [14,15]. Admittedly, injections into deeply located or small muscle groups represent a more precarious context for non-targeted BoNT administration [16,17,18]. This risk cannot be completely eliminated, though, even in large superficial muscles [19,20,21]. Several methods, such as imaging or electrophysiological techniques, can be used as an adjunct to facilitate target identification and increase the accuracy of needle placement [22,23,24,25]. The most commonly used techniques for muscle targeting comprise localization based on palpation and surface landmarks (manual needle placement or MNP), ultrasonography (US), electromyography (EMG) and electrical stimulation (ES) [22,23,24,25].

The necessity of guided BoNT injections has been subject to debate. Intuitively, a better therapeutic efficacy is anticipated with instrumented guidance compared to needle placement based on anatomical landmarks. Albeit logical, this has only been arbitrarily presumed by practitioners but has not been proven so far. In fact, there are reports suggesting an equivalence of the guided and non-guided approach, while expert opinions vary based on personal experiences in clinical practice [26,27,28,29]. Precedent reviews attempted to compare the effect of guided and non-guided injections on clinical outcome [30,31,32,33]. However, these reviews addressed this issue in the form of a critical appraisal and were restricted to a qualitative analysis, leading to less robust conclusions.

The role of instrumented guidance in BoNT injections and the superiority of one technique over the others in limb spasticity are disputable. To this, we conducted a network meta-analysis to address these questions by applying a quantitative analysis rather than a qualitative approach as already demonstrated in previous studies. Our objectives were first to investigate if guided BoNT injections lead to a better clinical outcome in adult patients with limb spasticity, and second to elucidate the hierarchy of guidance techniques including EMG, ES, MNP and US.

## 2. Materials and Methods

The network meta-analysis and systematic review was undertaken according to the PRISMA (Preferred Reporting Items for Systematic Reviews and Meta-Analyses) http://prisma-statement.org/ (accessed on 21 March 2023) and the PRISMA Extension Statement for Reporting of Systematic Reviews Incorporating Network Meta-analyses of Health Care Interventions [34]. This review was not registered.

### 2.1. Search Strategy

The literature research was performed using the electronic databases PubMed (MEDLINE), Scopus and CINAHL from database inception to December 2022. There were no language restrictions. A combination of keywords conjugated with the Boolean operators AND/OR as well as truncation and quotation were employed. A detailed description of the search strategy can be found in Figure 1 (in the Section 3). Moreover, unpublished studies were searched in clinicaltrials.gov and grey literature databases (Mednar, NTIS, Open Grey and York CRD). The reference lists of relevant articles were scrutinized to identify additional citations.

### 2.2. Eligibility Criteria

Studies eligible for inclusion had to be randomized or non-randomized clinical trials comparing two or more of the guidance techniques (EMG, ES, MNP and US) in adults with limb spasticity (upper or lower limb) treated with BoNT injections. Studies with BoNT injections in smooth muscles, non-limb (axial or head/neck) skeletal muscles, salivary glands or applied in the treatment of entities other than limb-spasticity were not deemed eligible. ES had to be used as a guidance technique for the injection and not as an adjunct therapy before or after BoNT injections to enhance therapeutic efficacy. The studies had to report the clinical outcome after the injection using the Modified Ashworth Scale (MAS). The time point of clinical evaluation was defined based on the observation by Chen et al. [35] that the clinical effects of BoNT peak at 3 to 4 weeks post-injection. However, as the time course of BoNT effects has not been clearly characterized, the range of acceptable follow-up was extended to 2 to 6 weeks post-injection. Eligible studies had to report the mean difference (MD) in MAS and the corresponding standard deviation (SD) between the pre- and post-injection clinical assessment. Alternatively, they had to provide sufficient data to calculate these estimates (MD and SD). Case series examining only one technique and case reports were not eligible. Systematic reviews, meta-analyses, technical reports and letters to the editor were also excluded.

### 2.3. Study Selection and Data Extraction

The study selection was performed by the first author (E.A.) and reviewed by the senior one (C.S.). Duplicate or irrelevant publications were excluded and the full text of relevant papers was reviewed. If multiple publications from the same author or the same institution were identified, the most recent paper was included. A final list of eligible records was created and the following data from each study were extracted: type of study, total number of patients, mean age, gender ratio (male/female), guidance techniques, underlying cause of spasticity, time point of clinical evaluation, assessment scales, MAS at baseline, type of BoNT used, doses and dilution, number of injection sites, muscles injected and adverse events during the procedure or in the follow-up period.

### 2.4. Assessment of Risk of Bias and Quality of Evidence

The risk of bias on the outcome level was evaluated using the revised Cochrane risk-of-bias tool for randomized studies (RoB 2 tool, version of 22 August 2019 for parallel-group trials/version of 18 March 2021 for crossover trials) [36] and the Newcastle–Ottawa Quality Assessment Scale for non-randomized studies [37]. Additionally, the certainty of evidence from the network meta-analysis was rated with the Grading of Recommendations Assessment, Development and Evaluation (GRADE) framework [38]. Individual appraisals were subsequently discussed, and the quality assessment was finalized.

### 2.5. Statistical Analysis

The principal summary measure was the difference in the means between the pre- and post-injection clinical evaluation in each arm of the study based on MAS. Additional summary measures were the guidance technique ranking and surface under the cumulative ranking curve (SUCRA) values. In studies reporting the median MAS and the interquartile range, the MD and SD estimates were extracted according to the methodology proposed by Wan et al. [39] and Cochrane Handbook [40]. As we compared multiple guidance techniques, we did not apply the classical pairwise meta-analysis but, instead, we implemented a network meta-analysis, which is an emerging statistical method combining evidence from direct and indirect comparisons [41,42,43]. The estimates were synthesized with a Bayesian network meta-analysis for continuous outcomes using the software MetaInsight V4.0.2 Beta [44]. Supplementarily, we performed a secondary analysis using the frequentist framework for continuous outcomes with the same software. The main analysis was conducted in the Bayesian framework, which is more flexible [45,46]. The Bayesian analysis does not rely solely on the observed data and on fixed population characteristics like the frequentist framework but treats both data and model parameters as random variables [45,46]. Nonetheless, for reasons of completeness, the analysis was additionally performed within the frequentist framework as well. The Meta-Insight tool utilizes the gemtc and netmeta packages of R for the Bayesian and frequentist framework, respectively. In both cases, we applied a random-effects model and for guidance technique ranking, smaller outcome measures (i.e., smaller MD before and after BoNT injection) were coded as undesirable. Forest plots showing the summary effects of all guidance techniques versus manual needle placement, ranking tables and SUCRA plots were generated.

Clinical and methodological heterogeneity were appraised both qualitatively and quantitatively by computing the Tau^2^, Chi^2^ and I^2^ estimates. The effect measure (MD in MAS before BoNT injection and at follow-up) was reported for each arm of each study. In other words, arm-level and not contrast-level summaries were used as input data. Therefore, correlations were not accounted for when including multi-arm trials [47]. The default prior distributions of the MetaInsight tool were used. To check the consistency assumption, the inconsistency test with the nodesplit model was undertaken. The model fit was evaluated by posterior mean of the residual deviance (Dbar), effective degrees of freedom (pD) and Deviance Information Criterion (DIC) estimates and visualized via residual deviance plots of the network meta-analysis model (NMA), unrelated mean effect model (UME), stem plots as well as leverage plots. Model convergence was assessed, based on the Brooks–Gelman–Rubin method after running multiple Markov Chain Monte Carlo (MCMC) simulations, and by visual inspection of Gelman convergence assessment plots. The transitivity assumption was tested by comparing clinical and methodological variables across the studies, such as mean age, mean MAS at baseline, number of injection sites, type of toxin, dilution, and dose. For the latter, we checked if the utilized doses were in accordance with existing dosing guidelines and manufacturer’s recommendations [1,48,49]. If the mean of the utilized dose was reported, then it had to be within the acceptable range. In case of different BoNT types, a conversion ratio of 3:1 (Abobotulinum A:Onabotulinum A or Incobotulinum A) was used to calculate the equivalent amounts [50]. Data entry was performed with Microsoft Excel 2016. Statistical analyses and visualization were performed using the MetaInsight software, the ggplot2 package of R and the Cochrane Review Manager (Revman) [51]. The level of significance was set at 0.05. Additional analyses included a multi-scale analysis, in which assessment scales other than the MAS were reported, and a sensitivity analysis by excluding studies with potential methodological flaws.

## 3. Results

### 3.1. Study Selection

A total of 1678 references were identified from the literature research. More specifically, we extracted 399 records from Pubmed, 812 from Scopus, 184 from CINAHL, 5 from clinicaltrials.gov, 232 from grey literature databases and 46 from the reference lists of retrieved articles. The titles were screened for relevance to our research question. Irrelevant or duplicate publications were excluded. Afterwards, 109 abstracts were further screened and 38 full-text reports were reviewed. All abstracts were in English and all studies whose full text was eligible for review were written in English as well, although there were no language restrictions in our literature search. Finally, six studies, all published in peer-reviewed journals, were deemed eligible [52,53,54,55,56,57]. No records from grey literature databases fulfilled the inclusion criteria. The flow chart with the process of study selection is depicted in Figure 1. In the vast majority of patients, the cause of spasticity was ischemic or hemorrhagic stroke.

### 3.2. Network Structure and Geometry

Six studies with 245 patients in total were included in the network meta-analysis. In particular, there were three two-arm and three multi-arm (three-arm) trials. The network graph resembles a rhombus with two diagonals and is shown in Figure 2. The graph comprises four nodes corresponding to the four guidance techniques. The size of each node reflects the number of patients assigned to each guidance technique (US: *n* = 69, ES: *n* = 73, EMG: *n* = 45, MNP: *n* = 58). Twelve patients in the US group were part of a crossover trial with EMG. There was direct evidence from three studies for MNP-US and ES-US, from two studies for MNP-EMG and MNP-ES and from one study for EMG-US as well as EMG-ES. MNP was used as a comparator in seven comparisons. Five comparisons were head-to-head comparisons between US, ES and EMG. All guidance techniques were adequately represented. As there are sufficient direct data, the network does not incorporate solely indirect comparisons and is well connected.

### 3.3. Characteristics and Quality Assessment of Individual Studies

From the included studies, five were randomized trials, among which four had a parallel group [54,55,56,57] and one a crossover design [53]. One study was a non-randomized trial [52]. The characteristics of each study are charted in Table 1 and a detailed description of the injected muscles as well as the dose/dose range for each muscle can be found in Appendix A. Serious adverse effects were not reported in any study. Only in the study of Ploumis et al. [55] did local reactions in the form of erythema appear in two patients (one in the EMG and one in the MNP group). The overall quality of the included studies was satisfactory. Three out of five randomized trials [54,55,56] had a low risk of bias, whereas in two [53,57] there were some concerns as assessed with the RoB2 tool. No study had a high risk of bias. One study [52] was non-randomized and had a high quality as appraised with the Newcastle–Ottawa scale. Additionally, the certainty of evidence was evaluated as moderate to high based on the GRADE framework. The detailed risk of bias assessment and the GRADE assessment are provided in Appendix A. Given the small number of studies (*n* = 6), the construction of a funnel plot and implementation of asymmetry tests were deemed inappropriate [58]. However, we had searched both for unpublished and published studies and we did not identify any unpublished reports that were eligible. The possibility for publication bias is rather low.

### 3.4. Bayesian Network Meta-Analysis Results

The Bayesian network meta-analysis revealed the hierarchy of guidance techniques in BoNT injections for limb spasticity. In particular, it was shown that US is the best among all techniques. ES was ranked second, EMG third and MNP fourth. It should be highlighted that although US was ranked first, the difference between US and ES was subtle, such that they could be regarded as almost equal. It is apparent that guided injections are superior to the non-guided ones. All three guided approaches lead to a better clinical outcome after BoNT treatment compared to needle placement entirely based on anatomical landmarks. These results are visualized in Figure 3 and Figure 4.

### 3.5. Assessment of Heterogeneity, Consistency and Transitivity

To measure heterogeneity, the Tau^2^, Chi^2^ and I^2^ statistics were initially computed. Evidence for non-substantial heterogeneity was provided quantitatively (Tau^2^ = 0.14, Chi^2^ = 15.47 with *p* = 0.05, I^2^ = 48%). Moreover, heterogeneity and transitivity were assessed qualitatively by scrutinizing clinical and methodological characteristics across the studies (Table 1 and Appendix A). Overall, there were no major discrepancies that could hamper the extraction of reliable conclusions. Minor deviations that were identified did not undermine the validity of the network meta-analysis. Special attention was paid to dosages, which had to be within an acceptable range as defined by consensus guidelines and manufacturers’ recommendations [1,48,49]. Regarding consistency, in line with the qualitative appraisal, a significant disagreement between direct and indirect estimates was not identified and in all comparisons the *p*-value was greater than 0.05 (Appendix A).

### 3.6. Assessment of Model Convergence and Measures of Fit

The computed measures of fit (Dbar = 10.34, pD = 9.91, DIC = 20.25) as well as the visualization tools indicated a good model performance. Between-study standard deviation was 0.07 (95% CI: 0–0.4). The NMA/UME residual deviance plot, stem plot and leverage plot are depicted in Figure 5. By inspection of these plots, it can be deduced that all study arms are characterized by a small posterior residual deviance, thus contributing to a better model fit. No poorly fitting data points were identified. A detailed explanation is provided in the legend of Figure 5 [59,60]. One study arm (MNP group in Zeuner et al.’s study) had a very good fit to the NMA consistency model, a small residual deviance and a high leverage. Therefore, a sensitivity analysis was performed to check if exclusion of this arm altered the results (see below in Section 3.7).

Moreover, convergence of the MCMC chains was observed. In our model the Brooks–Gelman–Rubin statistic (scale reduction factor) was computed with four independent Markov chains and it was in all cases very close to 1, meaning that variance between and within the chains was almost equal. Convergence was achieved at 5000 iterations. For the parameters d.EMG.ES, d.EMG.US and d.EMG.MNP scale reduction was stable over time, but for the parameter d.sd (i.e., between-study variation) scale reduction was principally stable but exhibited minor fluctuations. The Gelman convergence assessment plots can be found in Figure 5d.

### 3.7. Additional Analyses

In all studies, the authors used multiple scales to evaluate spasticity or other aspects. To avoid missing information from scales other than MAS, we visualized these results with a heatmap as computational analyses were not feasible. In this analysis, only studies which performed between-group comparisons for the corresponding scale were included. The heatmap is depicted in Figure 6 and the scoring strategy is described in the legend. US had more red “hits” (superiority hits) than the other guidance techniques when other scales of spasticity were taken into consideration, which is in congruence with the findings of the network meta-analysis.

In our study, five randomized trials and one non-randomized trial were included. The non-randomized study [52] fulfilled the eligibility criteria and extraction of the parameters of interest was feasible. Moreover, one of the randomized studies had a crossover design [53], while the other four were parallel-group trials [54,55,56,57]. The study with the crossover design was also included, as a time period of at least 3 months before the next BoNT treatment was maintained to avoid a carryover effect. In the study by Mayer et al. [57], the original Ashworth scale was used. Since the summary measure was the difference between the means (pre- and post-injection) and not the mean estimates per se, the difference by one level between the original and the modified scale was counterbalanced, and the study was also included. Lastly, in the study by Ploumis et al. [55], patients over 16 years old were eligible. Still, it was not clear if any of the included patients were between 16 and 18 years old, because individual data of age were not provided and the mean age was reported on a group level (EMG group: mean age 40.27, SD 15.82 and MNP group: mean age 41.83, SD 17.8). As violation of transitivity was not likely, this study was also included in the network meta-analysis. In order to test if the inclusion of these studies altered the results, sensitivity analyses (each time there was exclusion of one of the studies by Turna et al., Zeuner et al., Mayer et al. and Ploumis et al.) were performed. In all cases, the hierarchy of the guidance techniques was not altered, and US was always ranked first. In fact, the difference between US and ES was rather more prominent when these analyses were performed. However, the model fit was poorer. All these analyses can be found in the Appendix A.

When we tested deviance, we found that one study arm (MNP group in Zeuner et al. study) differed from the others, because it had an obviously better fit to the NMA consistency model, a very small residual deviance and a high leverage. Consequently, to exclude the possibility that the good model fit overall was driven by this, we excluded the MNP arm while keeping the EMG and US arms. The Bayesian analysis was repeated. Again, the ranking of the guidance techniques did not change (Appendix A). As we did not identify substantial heterogeneity, a meta-regression analysis was not undertaken.

Lastly, for reasons of completeness and methodological rigor, we complemented the Bayesian analysis with a frequentist network meta-analysis. Here, US and ES appeared as equal, EMG was in the third place and MNP was in the last place of the ranking. The league table and the forest plot are provided in Figure 7. Although the numerical estimates point to equivalence of US and ES, the visual software output of the league table ranked US at the top of the diagonal, implying that there still might exist a subtle predominance of US.

## 4. Discussion

This study is the first to provide quantitative evidence for the superiority of guided BoNT injections compared to anatomic localization and for the hierarchy of guidance techniques in terms of clinical efficacy. Here, we showed that US is the best method to guide BoNT injections in limb spasticity, followed by ES and EMG. All three approaches were superior to manual needle placement based on surface anatomy with regard to the clinical outcome as assessed by MAS at 2 to 4 weeks after BoNT treatment of limb spasticity in adults.

US was ranked first in all analyses, although the difference between US and ES was minor. In the Bayesian framework, US was found to be slightly better than ES, but in the frequentist framework, the two techniques were equivalent. Whilst the computed statistical estimates were identical and no difference was detected on the numerical level, in the graphical output of the frequentist meta-analysis, US was also ranked first. US predominance withstood all sensitivity analyses, in which the difference between the two techniques was slightly more prominent. Model comparisons during the sensitivity analyses revealed a better model fit when all six studies were included.

Substantial heterogeneity across these six studies was not identified, neither in technical parameters (such as volume, dilution and injection sites) nor in clinical characteristics. This was also supported by quantitative measures of heterogeneity. Additionally, it should be noted that only in the studies by Zeuner et al. [53] and Ploumis et al. [55] were a variety of muscles injected. In the other studies, a very limited number of muscles were injected. To ensure transitivity, probable effect modifiers were critically appraised. The effect of some divergencies in individual studies, in terms of study design [52,53], utilized assessment scale [57] and age of eligible participants [55], was checked in the sensitivity analyses but no major alterations were detected. The amount of BoNT injected in muscles in all studies was in accordance with dosage recommendations [1,48,49]. Although the assumption of transitivity was principally met, a potential confounder is additional rehabilitation following BoNT treatment. In two studies [54,56], the patients did not undergo any rehabilitation after BoNT injections, whereas this restriction was not imposed in the other studies. Hence, it might be that transitivity was compromised. Nonetheless, this discrepancy should not cause serious concerns given that available evidence indicates no effect or only very slight improvement by combining rehabilitation with BoNT than BoNT alone [61,62]. Consistency is the statistical manifestation of transitivity [38] and in our study, inconsistency was not detected (Appendix A), although the absence of inconsistency alone does not automatically exclude intransitivity. Taken together, both the critical and statistical evaluation of our synthesized evidence assure the validity of our results.

The previous investigations were based on MAS using the mean difference between the pre- and post-injection MAS score on the arm level. Extraction of the mean difference was not possible in all studies due to the way results were reported and, accordingly, quantitative evidence synthesis was not feasible for scales other than MAS. Nevertheless, MAS has some limitations, including subjectivity, suboptimal control of the stimulus and questionable reliability in some muscle groups of the lower limb [63,64,65,66]. In general, MAS reliability and validity are adequate, especially when spasticity is evaluated by trained medical professionals as in the studies of our meta-analysis [67,68]. Yet, other scales such as the Modified Tardieu Scale (MTS) have been shown to have a better construct validity [69,70,71]. Despite any drawbacks, MAS is the most widely used assessment scale of spasticity in clinical practice and was the most commonly used scale by authors. To incorporate data from spasticity scales that could not be quantitatively synthesized, we conducted an additional qualitative analysis which took into account spasticity scales other than MAS as well as scales evaluating other aspects, such as functional independence in daily life. Again, US was found to be superior to the other injection techniques in terms of spasticity improvement, further corroborating the results of the quantitative analysis.

An alternative solution when dealing with multiple scales is the use of the standardized mean difference as a summary effect so that different scales measuring the same dimension in different studies are taken into account. This was not applicable in our study, since there were cases [54,56,57] in which spasticity was measured with two or even three different scales within the same study. In other words, several scales were used to measure the same construct (spasticity) in the same study and not across studies. The authors had stated that the motivation to use simultaneously different scales for spasticity was to account for limitations associated with each scale. Apart from that, other scales that were used measured items other than spasticity (e.g., quality of life), but computations within the framework of a network were not plausible due to insufficient data.

The advantage of guided against non-guided BoNT injections is often taken as a given, because it is intuitive to expect a better therapeutic efficacy by introducing an imaging or electrophysiological adjunct to traditional anatomical localization. To date, this has been supported solely by qualitative evidence synthesis methodologies [30,31,32,33]. Our study proves for the first time the superiority of guided BoNT injections by implementing a network meta-analysis and quantifies these effects for limb spasticity in adults. It should be noted that these results should not be imprudently generalized for other clinical entities treated with BoNT or for pediatric populations. Our findings definitely provide meaningful insights and could imply a similar effect in other diseases, but this needs to be confirmed in future studies.

The network meta-analysis suggested that guidance of BoNT injections in spastic limbs is essential. The ensuing question was which method is the best to guide BoNT injections in adults with limb spasticity and our overall results demonstrated a superiority of US. ES was ranked second and EMG third. This hierarchy should not be viewed as a rigid construct, though. The difference between US and ES was minor and it could be argued that the two techniques are almost equal. Thus, it is important to take other factors into consideration for decision making. The choice of the adjunct method is influenced by the clinical setting (private practice or hospital), available equipment, injector experience, training as well as patient preferences. Importantly, we did not identify a higher number of adverse effects associated with the utilization of one technique.

US enables real-time visualization of needle placement, and as a result damage of critical neurovascular structures and accidental BoNT injection into contiguous muscles can be avoided [23]. Moreover, US offers flexibility in maneuvers. For instance, if spread of the toxin is noticed during the injection, the injector can alter the position of the needle to avoid further diffusion and inadvertent weakness of non-spastic muscles. At the same time, anatomical variations or spasticity-induced alterations in muscle trophicity can be discerned and needle placement can be adjusted accordingly [26]. US is particularly useful when a pinpoint accuracy is warranted, such as in fascicle localization within the muscle or in small overlapping muscles. This possibility for dynamic visualization is an asset. The strength of US lies in the ability to provide a panoramic view of the injection area and information regarding muscle architecture, thus allowing safe and accurate manipulations. Further, US causes less discomfort and pain to the patient compared to electrophysiological approaches, during which the needle remains within the muscle for a longer time and more needle sticks are required.

On the other hand, both ES and EMG allow a more precise identification of motor endplates, which leads to a more effective blockade of neurotransmission [72,73]. In turn, a lower dosage is required [74,75], which is linked to lower probability of spread into adjacent muscles (i.e., inadvertent weakness) and lower costs. However, in the included studies there was not any tailoring of the dose according to the guidance technique, and conventional injection points were used. US cannot quantify electrical activity and as a result motor endplates cannot be localized. This drawback is important, since motor endplate injections represent a promising strategy to achieve clinical efficacy with a lower BoNT dosage. US is also compromised by high equipment costs as well as the need for adequate training and experience of the injector. EMG and ES are technically less challenging. In EMG, acoustic or visual feedback allows quantification of muscle spasticity, and the needle can be adjusted until the optimal point (as close as possible to the motor endplate) is detected. The closer the motor endplate is, the more intense the EMG signal is. Proximity to the motor endplate can also be estimated by ES. In this case, the closer the motor endplate, the lower the required electrical intensity is to maintain muscle contraction. Both EMG and ES entail dynamic components in the sense that the injector can reposition the needle until the best possible point for the injection is identified. This acquires special significance, because different muscles have different motor endplate zones [75,76]. Despite that, electrically quiescent structures cannot be appraised.

It should be highlighted that in the included studies it was stated that the injector was experienced. The years of experience were also available in three studies (more than 5 years [56], more than 6 years [54] and more than 16 years [52]). Admittedly, the experience of the physician is fundamental for clinical efficacy. Given that in our meta-analysis the BoNT injections were performed by experienced injectors, our findings are applicable only for trained practitioners with experience, as it is highly possible that the hierarchy is quite different for less-experienced injectors. Albeit pivotal, the experience of the injector and its influence on the outcome has not been adequately investigated in studies with clinical populations. This aspect has been addressed in two cadaveric studies [21,77]. It seems that when it comes to non-guided injections, the accuracy of needle placement does not differ significantly between experienced and non-experienced injectors, but for guided injections, it still remains obscure whether there is a difference in accuracy of needle trajectory between experienced and non-experienced injectors. Another cadaveric study compared accuracy between MNP and US-guided injection in neck muscles [78]. Both in guided and non-guided injections there was no statistically significant difference based on the level of injector experience. Experienced injectors always performed a bit better than the less-experienced ones in both cases. Interestingly, the difference when US was applied was smaller, meaning that US actually helped the less experienced physician to perform better. In this study, though, cervical and not limb muscles were injected. Regardless, these implications warrant further investigation through a larger number of practitioners with a different training level.

The results of this network meta-analysis are in line with the conclusions of previous narrative and systematic reviews that implemented a qualitative approach [30,32,33]. Walker et al. [32] and Chan et al. [33] presented the results of individual studies and suggested that guidance techniques lead to a better outcome than MNP alone. However, general conclusions about the superiority of one or more guidance techniques could not be drawn. Grigoriu et al. [30] analyzed pertinent clinical studies with adult and pediatric populations in a more systematic manner and recommended instrumented guidance during BoNT injections. The authors noted that there were no major differences between US and ES, although US seemed to be superior to ES for spastic equinus in adults with stroke. The study results were not conclusive for EMG, though. We also confirmed in a quantitative manner that guided injections are better than the non-guided ones. Moreover, we showed that EMG is inferior to US and ES, which are comparable although US is slightly superior. It should be pointed out that previous reviews examined the topic for several muscle groups and clinical entities, while we focused on adult limb spasticity. As both superficial and deep muscles were injected in the included studies, it is not possible to draw conclusions about the usefulness of each technique depending on the layer of the muscle. The need to tailor the guidance modalities according to the underlying disease, anatomic compartment, muscle groups or even individual muscles should be investigated in future clinical trials.

## 5. Recommendations for Clinicians

The findings of our meta-analysis should be appropriately contextualized. Certainly, our results should encourage clinicians to use imaging or electrophysiological techniques to guide BoNT injections in adults with limb spasticity, as this leads to a better clinical outcome. EMG was found to be inferior to ES. If someone adopts the motor endplate strategy and needs to opt for an electrophysiological method, then ES might be preferable, provided that the discomfort caused to the patient is acceptable. The difference between US and ES was minor and therefore, other aspects should be considered. The availability of the equipment plays a significant role. If, for example, a practitioner in a private practice is trained and already has a US machine which is also used for other purposes, then US might be preferred because of its high potential for a good clinical outcome and the fact that less discomfort is caused. The latter aspect should not be understated, as it is crucial to ensure the compliance of patients to treatment sessions. US might also be preferable in academic hospitals, where patients with limb spasticity as a manifestation of less-common diseases are treated. In this instance, US offers the advantage for real-time visualization and corresponding needle adjustments, which is critical when altered anatomical configurations are expected. For a less-experienced physician, US might be challenging at first while ES might be proper for this initial stage either alone or simultaneously with US until acquaintance with US is achieved.

## 6. Future Perspectives

It becomes apparent that BoNT injections can be optimized through guidance techniques, either imaging or electrophysiological. Our study suggests that US and ES lead to a better clinical outcome than EMG and MNP, with a slight superiority of US. In the future, clinicians should try to introduce a guidance technique in their daily practice depending on their working setting. In that sense, they should also pursue relevant training opportunities. Future clinical trials should also focus on comparisons between US and ES in larger study populations, so that more robust evidence can be provided. Combinations of different techniques on the same patient are also an interesting perspective, since in this way the limitations of one technique can be counterbalanced by the advantages of another. Novel technologies and approaches other than the classical ones, such as fluoroscopy [79], TC99m-MIBI-SPECT-CT [80] and 3D innervation zone imaging [81], should also be researched.

## 7. Limitations

Our study has some limitations. Only six studies with 245 patients were eligible for inclusion. The number of studies is low, but the quality in all of them was satisfactory and, overall, the certainty of the evidence based on the GRADE approach was good. Additionally, our results are restricted to limb spasticity and cannot be generalized for spasticity in axial as well as head/neck muscles or other entities such as dystonia, in which BoNT injections are also applied [27,28,29]. It should also be noted that the authors of the included studies performed manual needle placement based on conventional injection points rather than injections based on intramuscular patterns of the nerve distribution as suggested for some limb muscles [82,83]. Further, network meta-analyses have some limitations, including a higher heterogeneity compared to pairwise meta-analyses as well as compromise of consistency, if direct and indirect evidence deviate [84,85]. Yet, in our study, no substantial heterogeneity or inconsistency were observed. With regard to the utilized dose, in two studies [52,55] the exact dose or dose range were not reported. As a result, we had to approximate the dose for each muscle based on the total dose and other available information, so there might be some minor deviations. Finally, our results were based on a clinical evaluation of spasticity according to MAS, which is susceptible to subjectivity. An emerging concept to overcome this limitation in future studies is the use of sono-elastography to objectively evaluate spasticity after BoNT injections [86,87,88]. This approach has not been established yet and, hence, previous studies including those in our network meta-analysis did not utilize this method. However, sono-elastography offers a new perspective towards a more objective assessment of spasticity after BoNT injections.

## 8. Conclusions

In conclusion, instrumented guidance during BoNT injections performed by experienced practitioners leads to a better clinical outcome within the first month in adults with limb spasticity when compared to non-guided injections. In this context, US is superior to the other guidance techniques and the exact hierarchy comprises US on the first level, ES on the second, EMG on the third and MNP on the last. The difference between US and ES is minor and these two techniques could be regarded as equal. Thus, appropriate contextualization according to clinical setting, injector experience, available equipment and patient preferences is essential. The focus of future clinical trials should be moved to the comparison of US with ES to shed more light on which modality is superior.

## Figures and Tables

**Figure 1 toxins-15-00256-f001:**
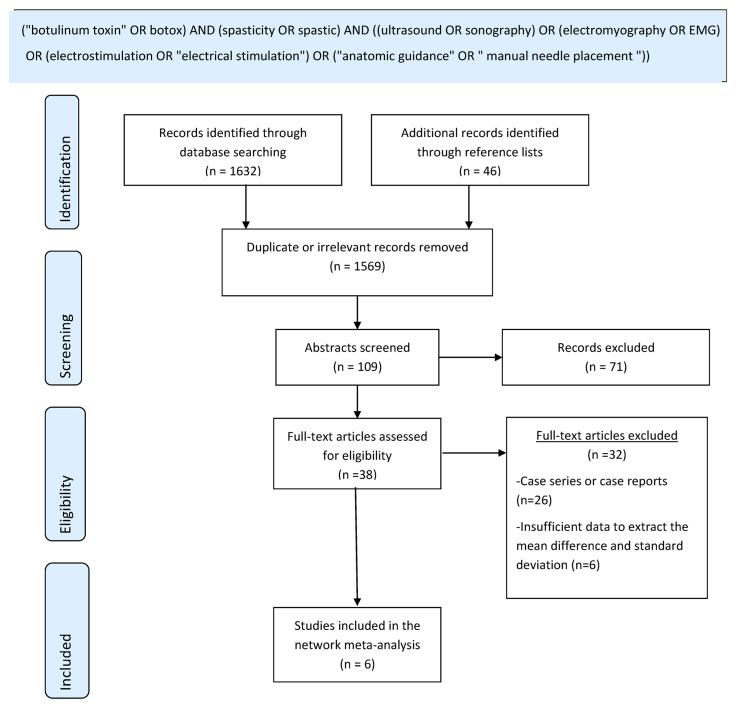
Flow diagram for study selection.

**Figure 2 toxins-15-00256-f002:**
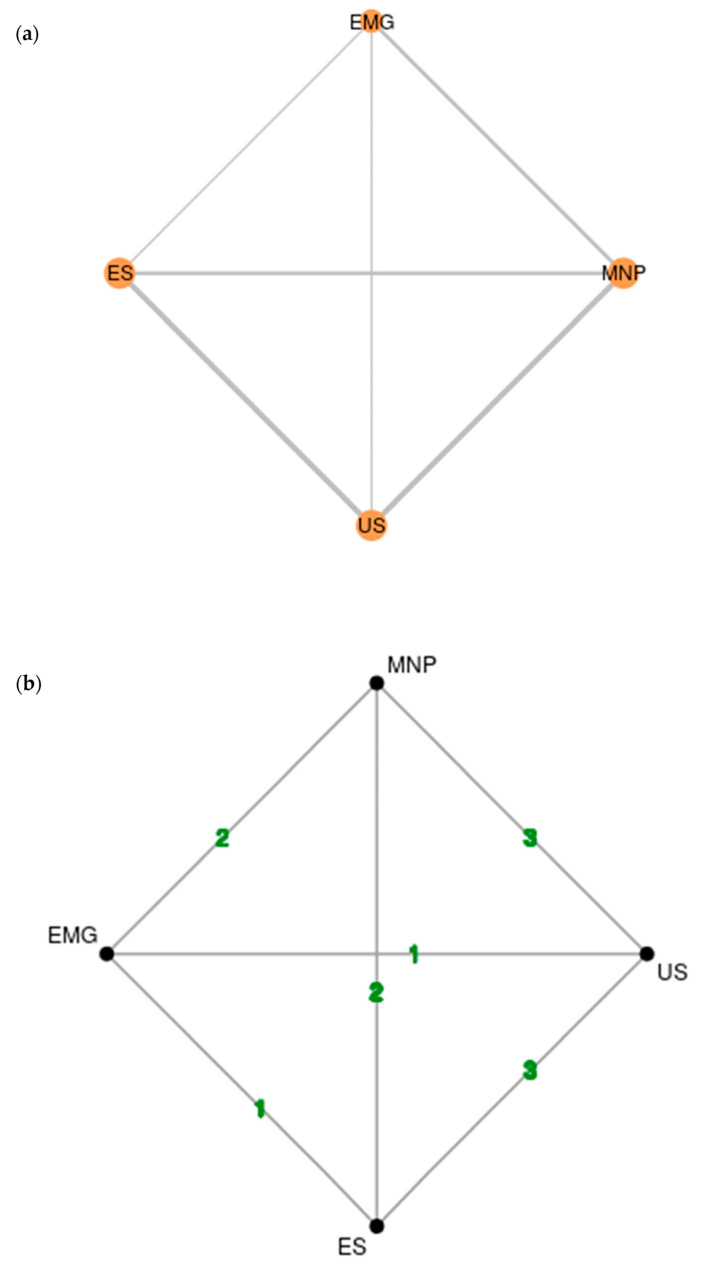
(**a**) Rhombus-shaped network plot with four nodes corresponding to one guidance technique. The size of each node depends on the number of patients assigned to this technique. (**b**) Weighted network graph showing the number of studies in each edge. EMG: electromyography, ES: electrical stimulation, MNP: manual needle placement, US: ultrasound.

**Figure 3 toxins-15-00256-f003:**
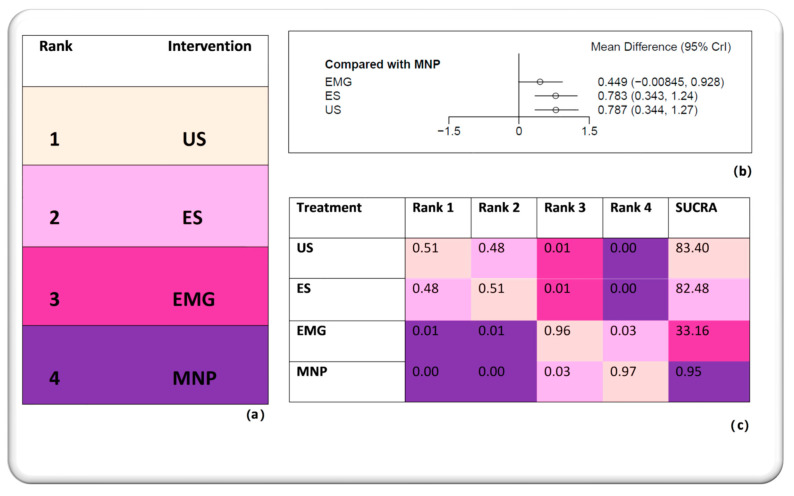
(**a**) The hierarchy of guidance techniques as shown in the Bayesian network meta-analysis. (**b**) Forest plot depicting the effect estimate with the corresponding confidence intervals for each guidance technique when compared to the non-guided approach. The guidance techniques are ranked from the worst at the top to the best at the bottom of the plot. (**c**) Ranking table showing the probability of each technique being ranked as first, second, third and fourth as well as the SUCRA values. Higher SUCRA values indicate a higher probability to be ranked as the best technique. EMG: electromyography, ES: electrical stimulation, MNP: manual needle placement, SUCRA: surface under the cumulative ranking curve, US: ultrasound, 95% CrI: credible interval.

**Figure 4 toxins-15-00256-f004:**
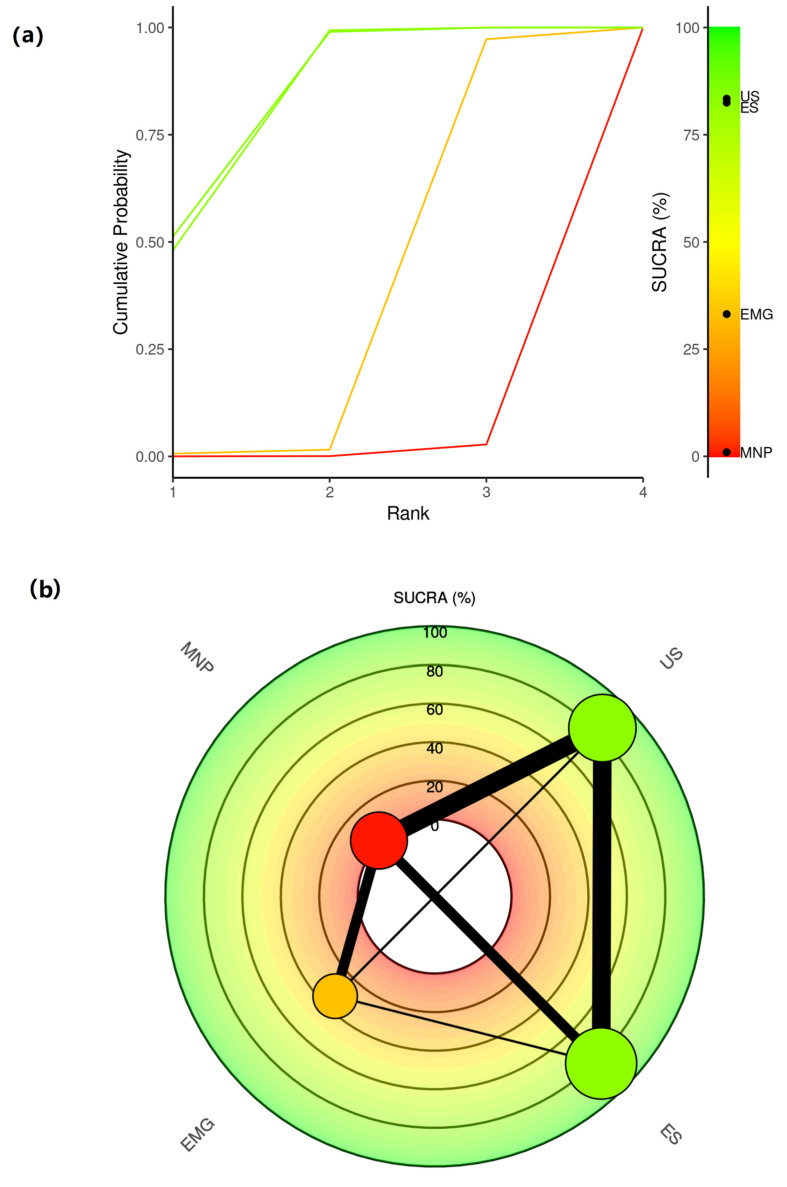
Main ranking results with two different plot types: (**a**) The Litmus Rank-O-Gram shows the cumulative probability for each guidance technique to be ranked as first, second, third and fourth alongside with the SUCRA values. (**b**) Radial SUCRA plot displaying SUCRA values for each treatment radially with a network diagram of evidence overlaid. EMG: electromyography, ES: electrical stimulation, MNP: manual needle placement, SUCRA: surface under the cumulative ranking curve, US: ultrasound.

**Figure 5 toxins-15-00256-f005:**
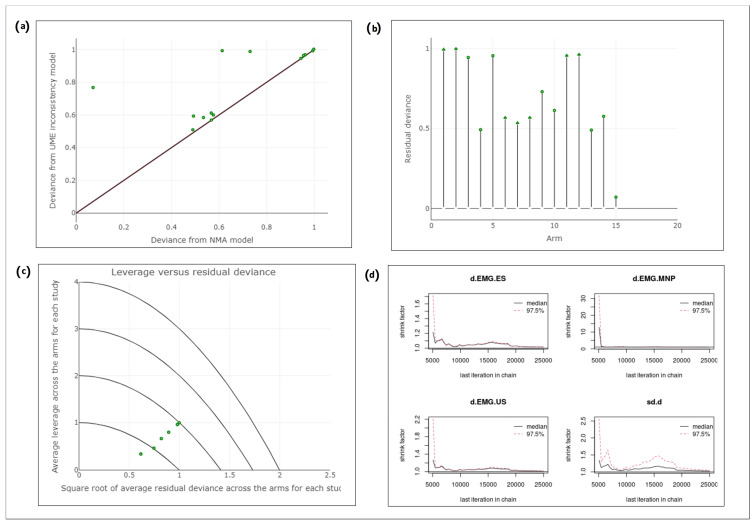
(**a**) The NMA/UME residual deviance plot displays the contribution of each data point (study arm) to the residual deviance for the NMA consistency and the UME inconsistency models. If the data points are located on the line of equality, there is no inconsistency, because the model fit does not improve if the UME inconsistency model is applied. Points below the equality line have a better fit for the UME inconsistency model. On the contrary, points above the equality line have a better fit for the NMA consistency model. Hence, data points above or on the equality line have a smaller residual deviance from the NMA consistency model and there is no proof of inconsistency. In our case, all study arms were above or on the equality line, meaning that there was not inconsistency. (**b**) The stem plot visualizes the posterior residual deviance of each study arm. The shorter the stem, the smaller the residual deviance and thus, the better the model fit. In total, there are 15 study arms and in all cases posterior residual deviance is lower than 1, pointing to a good model fit. (**c**) The leverage plot is used to evaluate the influence of each data point to the model fit and DIC. The average leverage is depicted on the y-axis and the residual deviance is on the x-axis. There are parabolas characterized by a number, represented by c. Points that lie on each parabola or in-between contribute an amount of c to the model estimation. Points lying outside the curve with c = 3 contribute to a poor model fit. In our study, all study arms lie below the parabola with c = 1.5 and contribute to a good model fitness. (**d**) Gelman convergence assessment plots for the parameters d.EMG.ES, d.EMG.US and d.EMG.MNP and d.sd. Four Markov chains were used to compute the scale reduction factor. In each case, there were 25,000 iterations and convergence was achieved at 5000 iterations approximately. Variance between and within the chains was almost identical, in other words there was a satisfactory convergence of the Markov chains. The chain steps were stable over the time apart from the d.sd parameter in which they exhibited small divergence. Abbreviations: EMG: electromyography, ES: electrical stimulation, MNP: manual needle placement, NMA: network meta-analysis, sd: standard deviation, UME: unrelated mean effect, US: ultrasound.

**Figure 6 toxins-15-00256-f006:**
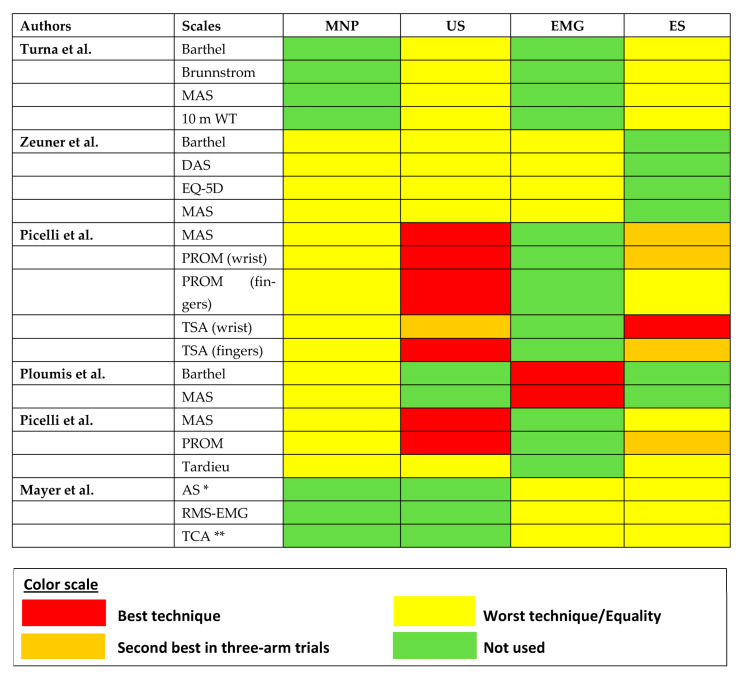
Heatmap showing the results of between-group comparisons in each study across all utilized assessment scales. Superiority of a guidance technique is labeled as red. Orange codifies the second-best technique in three-arm trials. Yellow designates a technique which is the worst in threarm or two-arm studies. When both techniques are colored with yellow, they are equal and there was no difference in their efficacy. Techniques that were not evaluated in this study are marked with green. It becomes apparent that US has more superiority “hits” across different spasticity scales compared to the other guidance techniques but not in non-spasticity scales. Abbreviations: AS: Ashworth Scale, DAS: Disability Assessment Scale, EMG: electromyography, EQ-5D: Quality-of-Life Scale, ES: electrical stimulation, MAS: Modified Ashworth Scale, MNP: manual needle placement, PROM: ankle passive dorsiflexion range of motion, RMS-EMG: root mean square of surface electromyographic activity during the Ashworth maneuver, TCA: Tardieu catch angle, TSA: Tardieu spasticity angle, US: ultrasound, 10 mWT: 10 m walking test. * In this study the original Ashworth scale was used. Since the summary measure was the difference between the means pre- and post-injection and not the mean estimates per se, the difference by one level between the original and the modified scale was counterbalanced. ** TCA (Tardieu catch angle) is the same as TSA (Tardieu spasticity angle) [52,53,54,55,56,57].

**Figure 7 toxins-15-00256-f007:**
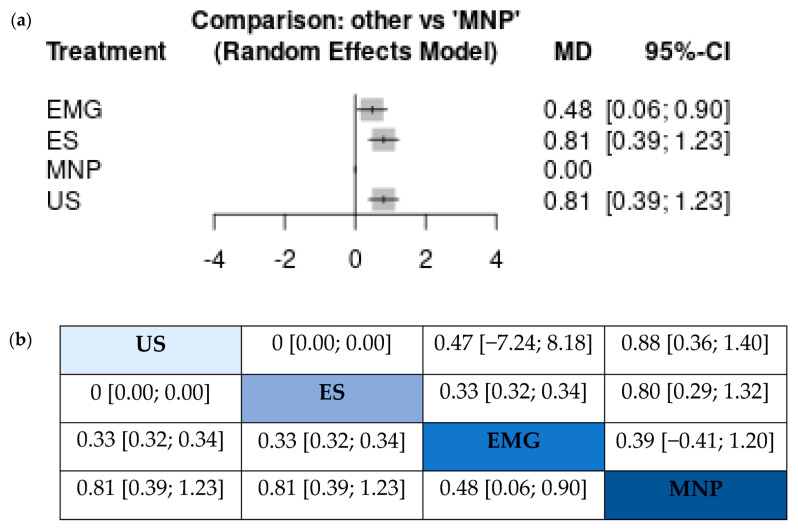
Frequentist network meta-analysis results: (**a**) Forest plot showing the output of comparisons between each guidance technique and non-guided injections. Ultrasound and electrostimulation appear to be equivalent. (**b**) League table showing the results of pairwise comparisons as mean difference and the corresponding credible intervals. The four techniques are ranked from the top to the bottom of the diagonal (from the best to the worst). Of note, ultrasound is again depicted at the top of the diagonal, although the statistical estimates within the frequentist framework point to an equality of ultrasound and electrical stimulation. Abbreviations: EMG: electromyography, ES: electrical stimulation, MNP: manual needle placement, US: ultrasound.

**Table 1 toxins-15-00256-t001:** Overview of clinical and methodological characteristics for the included studies. Study design, number of patients, gender ratio, cause of spasticity, utilized assessment scales, time of post-injection clinical evaluation, compared guidance techniques, botulinum toxin type, dilution and number of injection sites in each muscle are presented. Abbreviations: AS: Ashworth Scale, DAS: Disability Assessment Scale, EMG: electromyography, EQ-5D: Quality-of-Life Scale, ES: electrical stimulation, f: female, m: male, MAS: Modified Ashworth Scale, MNP: manual needle placement, n: number, NA: not available, PROM: ankle passive dorsiflexion range of motion, RCT: Randomized controlled trial, RT: Randomized trial, RMS-EMG: root mean square of surface electromyographic activity during the Ashworth maneuver, TCA: Tardieu catch angle, TSA: Tardieu spasticity angle, U: units, US: ultrasound, w: week, 10 mWT: 10 m walking test. * In this study the original Ashworth scale was used. Since the summary measure was the difference between the means and not the mean estimates per se, the difference by one level between the original and the modified scale was counterbalanced. ** TCA (Tardieu catch angle) is the same as TSA (Tardieu spasticity angle).

Authors (Year)	Type of Study	No of Patients (m/f)	Cause of Spasticity	Assessment Scales	Post-Injection Clinical Evaluation	Guidance Technique (n)	Type of Toxin (Brand Name)	Dilution	Injection Sites/Muscle (n)
Turna et al. (2018) [52]	Prospective cohort	40 (23/17)	Ischemic or hemorrhagic stroke	Brunnstrom stage, Barthel Index, MAS, 10 mWT	2 w 3 m	ES (20) US (20)	Abobotulinum toxin A (Dysport^®^) Onabotulinum toxin A (Botox^®^)	1000 U in 2.5 mL NaCL 300 U in 2 mL NaCL	NA
Zeuner et al. (2016) [53]	Crossover RCT	23 (10/13)	Ischemic or hemorrhagic stroke	Barthel Index, DAS, EQ-5D, MAS	4 w	EMG (12) MNP (11) US (12)	Onabotulinum toxin A (Botox^®^)	100 U in 2mL NaCL	NA
Picelli et al. (2014) [54]	Parallel-group RCT	60 (32/28)	Ischemic or hemorrhagic stroke	MAS, PROM, TSA	4 w	ES (20) MNP (20) US (20)	Abobotulinum toxin A (Dysport^®^)	500 U in 2 mL NaCL	1
Ploumis et al. (2013) [55]	Parallel-group RCT	27 (7/20)	Stroke, traumatic brain injury, spinal cord injury, cerebral palsy, hypoxic encephalopathy	Barthel Index, MAS	3 w 3 m	EMG (15) MNP (12)	Onabotulinum toxin A (Botox^®^)	100 U in 1 mL NaCL	1–2
Picelli et al. (2012) [56]	Parallel-group RCT	47 (31/16)	Ischemic or hemorrhagic stroke	MAS, PROM, TSA	4 w	ES (15) MNP (15) US (17)	Onabotulinum toxin A (Botox^®^)	100 U in 2 mL NaCL	2
Mayer et al. (2008) [57]	Parallel-group RT	36 (18/18) (elbows)	Stroke, traumatic brain injury, hypoxic encephalopathy	AS *, RMS-EMG, TCA **	3 w	EMG (18) ES (18)	Onabotulinum toxin A (Botox^®^)	60 U in 2.4 mL NaCL and 30 U in 1.2 mL NaCL	1 site in motor point injections 4 (biceps) and 2 (brachioradialis)

## Data Availability

The data that support the findings of this study are available from the corresponding author upon reasonable request.

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
