# Peer review of "A Bayesian Network Meta-Analysis and Systematic Review of Guidance Techniques in Botulinum Toxin Injections and Their Hierarchy in the Treatment of Limb Spasticity"

_toxins, 2023, doi:10.3390/toxins15040256_

Round 1

Reviewer 1 Report

This study investigated whether guided botulinum neurotoxin (BoNT) injections lead to better clinical outcomes than non-guided injections for adults with limb spasticity. The study also aimed to determine the hierarchy of common guidance techniques including electromyography, electrostimulation, manual needle placement, and ultrasound. The study conducted a Bayesian network meta-analysis with 245 patients and found that guided BoNT injections based on ultrasound and electrostimulation performed by experienced practitioners lead to better clinical outcomes within the first month post-injection. The hierarchy of guidance techniques was found to be ultrasound on the first level, electrostimulation on the second, electromyography on the third, and manual needle placement on the last level. The study suggested that appropriate contextualization is essential for decision-making and that large-scale trials should shed more light on which modality is superior.

In the introduction, in sentence 31, it should be "subsequent" instead of "subsequently". 

A precise anatomical localization maximizes the amount of BoNT administered within the target structure, at the same time minimizing unintended weakness of adjacent muscles due to toxin spread. Cite a reference of these two articles “Effective botulinum neurotoxin injection in treating iliopsoas spasticity” “Anatomical locations of the motor endplates of sartorius muscle for botulinum toxin injections in treatment of muscle spasticity”.

In line 136, you could consider using a semicolon instead of a comma to separate the two independent clauses: "Clinical and methodological heterogeneity were appraised both qualitatively and quantitatively by computing the Tau2; Chi2 and I2 estimates."

In line 139, you could consider rephrasing "Hence, there was no need to account for correlations for including multi-arm trials" to something like "Therefore, correlations were not accounted for when including multi-arm trials."

In line 145, you could consider adding a comma after "model convergence was assessed": "Model convergence was assessed, based on the Brooks-Gelman-Rubin method after running multiple Markov Chain Monte Carlo simulations, and by visual inspection of Gelman convergence assessment plots."

In line 149, you could consider adding a comma after "methodological variables across the studies": "The transitivity assumption was tested by comparing clinical and methodological variables across the studies, such as mean age, mean MAS at baseline, number of injection sites, type of toxin, dilution, and dose."

In line 157, you could consider adding a comma after "the ggplot2 package of R": "using the MetaInsight software, the ggplot2 package of R, and the Cochrane Review Manager (Revman) [29]."

In the discussion, please added the limitation of the study that it had conventional injection point rather than injection based on intramuscular patterns of the nerve distribution. “Distribution of the intramuscular innervation of the brachial triceps: clinical importance in the treatment of spasticity with botulinum neurotoxin” 

I would like to give major revision and want to have reviewed again after revised version

Reviewer 2 Report

The authors did a Bayesian network meta-analysis on guidance techniques in botulinum toxin injection in the treatment of limb spasticity and concluded that ultrasound-guided injection performed slightly better. There are a few concerns about the analysis design and data extraction. Authors used Bayesian analysis, they should provide the justification why using Bayesian meta-analysis over the classical meta-analysis, and the potential limitations with Bayesian analysis. The conclusion is the US-guided injection is slightly better than electrical stimulation or electromyography-guided injection. However, as authors pointed out in the discussion, US-guided injection used higher doses of toxin. Can they do a compounding analysis on the dose and see if the US-guided injection is still better? On the data extraction, authors seem not to extract the dose information correctly. Ref. 30 used 1000 U of Abobotulinum toxin A (Dysport), ref 31 different muscles used different doses with different total doses for different groups; ref 32, different muscles used different doses; ref 35, different muscles used different doses with total dose of 90 units. Those dose effects may play roles in the outcomes, and when compounding to the final analysis, the conclusion may change. The doi numbers in the references are also not correct for some references. Authors need to correct them.

Reviewer 3 Report

In this systematic review the authors provided quantitative evidence for superiority of guided BoNT injections compared to anatomic localization and for the hierarchy of guidance techniques in terms of clinical efficacy.

The idea of this study is interesting; unfortunately, this manuscript needs some improvements and corrections before publishing may be possible.

General points:

Title page: please correct: Systematic Review and Meta-Analysis

Please add a list of abbreviations before References section to your manuscript.

Please add a Future perspectives section to your manuscript.

Please add the recommendations for clinician for BoNT injections in the treatment of limb spasticity.

Special points:

Keywords: please add to keywords: injections.

Introduction

Lines29-35: please add multiple references at the end of each these sentences.

Lines 36-49: please add multiple references at the end of each these sentences.

Materials and Methods

Lines 73-74: how is it possible: there were no language restrictions. Please say, which languages you included in your analysis?

Results

Lines 215-218: please add multiple references at the end of each these sentences.

 Line 227: please add the appropriate references for Bayesian Network.

Discussion

Lines 583-584: please describe all these studies exactly.

Round 2

Reviewer 1 Report

Everything the authors have answered are clear and proficient.

I would like to accept in current form.

Reviewer 2 Report

The revision is fine. It can be accepted after the editorial check, especially the references and the doi numbers for each reference.

Reviewer 3 Report

Thak you for all corrections.